# Effect of ultrasound-guided transverse abdominal plane block on neutrophil-to-lymphocyte ratio, platelet-to-lymphocyte ratio, and systemic immune inflammation index in patients undergoing radical resection of endometrial carcinoma

**Changxu Wang[1]☯, Shiwen Fan[1,2]☯, Dengfeng Gu[1], Jiaojiao Deng[1], Baobao Ma[1], Liping Xie[1‡]\*, Hong Zhang◉[1‡]\***

**1** Department of Anesthesiology, First Affiliated Hospital, Shihezi University, Shihezi, China, **2** Department of Anesthesiology, Union Hospital, Tongji Medical College, Huazhong University of Science and Technology, Wuhan, China

☯ These authors contributed equally to this work.
‡ LX and HZ also contributed equally to this work.
\* 1579927013@qq.com (HZ); xielipingmazui@163.com (LX)

## Abstract

### Objective

The purpose of this trial was to explore the effects of the ultrasound-guided transverse abdominal plane block (TAPB) on the systemic immune-inflammatory index (SII), peripheral blood neutrophil to lymphocyte ratio (NLR), platelet to lymphocyte ratio (PLR) in patients undergoing radical resection of endometrial carcinoma.

### Methods

This trail was registered in the Chinese Clinical Trial Registry (ChiCTR2300072186, www.chictr.org/; approval date: 2023-06-06). In the study, a total of 90 patients who were scheduled for radical resection of endometrial carcinoma were selected, and they were randomized to receive ultrasound-guided TAPB combined with general anesthesia (GA) or either GA. The primary outcomes were the values of NLR、PLR and SII which were obtained at postoperative 24 hours and 72 hours. Other observational indicators included: the counts of neutrophil, lymphocyte, and platelet; the numbers of effective press of the analgesic pump; postoperative pain intensity; remifentanil consumption; and adverse reactions.

### Results

The values of preoperative peripheral blood neutrophil, platelet, lymphocyte, NLR, PLR, and SII did not differ between the two groups (*P*>0.05). The TAP+GA group exhibited

**Data Availability Statement:** All relevant data are within the paper and its Supporting Information files.

**Funding:** The author(s) received no specific funding for this work.

**Competing interests:** The authors have declared that no competing interests exist.

significantly reduced levels of neutrophil, NLR, and SII at 24 and 72 hours post-surgery than the GA group (P<0.05). However, there were no significant differences in the values of PLR between the two groups (*P*>0.05). Compared with the GA group, the VAS scores at 6 hours, 12 hours, and 24 hours after surgery in the TAP+GA group were significantly decreased, and the intraoperative consumption of remifentanil and the numbers of postoperative analgesic pump presses were significantly reduced (*P*<0.05). Moreover, the incidence of postoperative nausea and vomiting was reduced considerably in the TAP+GA group (*P*<0.05).

## Conclusions

Ultrasound-guided TAPB can effectively lower the values of postoperative neutrophil, NLR, and SII, improve postoperative pain intensity, decrease opioid consumption, and reduce the incidence of postoperative nausea and vomiting.

## Introduction

Endometrial cancer is one of the most common malignant tumors of the female reproductive system [1], the significant increase in its incidence [2, 3] and the devastating prognosis after metastasis and recurrence [4] are currently major concerns. Surgery stands as the primary method for treating endometrial cancer, multiple factors [5–8] such as anesthesia, immune system, blood transfusion, surgery, and postoperative pain can promote or hinder the metastasis of residual tumor cells in the perioperative period. Kim et al. found that the selection of different anesthesia methods during the perioperative period can directly or indirectly impact the prognosis of cancer patients [9, 10]. Consequently, exploring additional perioperative interventions' effects on the recurrence and metastasis of endometrial cancer will be aid in the treatment of cancer patients.

Transverse abdominal muscle plane block (TAPB) is a nerve block method, that is mostly used to relieve postoperative pain response in patients undergoing lower abdominal surgery, and it has also been found to have a positive effect on inhibiting inflammatory response and improving postoperative immune function [11].

Cancer-associated inflammatory response and immunology are increasingly viewed as crucial mechanisms in tumor formation, with mounting evidence that systemic inflammation may play a key role in tumor growth, progression, invasion, and metastasis [12]. Excessive stress on the body due to surgical trauma, extensive opioid use, and postoperative pain, all of which can suppress cellular immune function [13]. How to optimize perioperative interventions, mitigate the above adverse effects, enhance the postoperative immune function of patients, and thus improve the prognosis of tumor patients is an issue that anesthesiologists must also focus on under the ERAS philosophy. Peripheral blood inflammatory cells such as neutrophils, lymphocytes, and platelets are unstable due to individual differences, rather, the NLR, PLR, and SII, which are calculated from these indicators, present a relatively stable state and have high predictive value for the prognosis of cancer patients [14–16]. The clinical significance of these blood-derived inflammatory indicators in female malignant tumors is increasingly being recognized, and there have been studies showing that high preoperative and postoperative NLR is associated with poor prognosis in endometrial cancer patients [17]. It has also been shown that SII is suitable for predicting the short- and long-term prognosis of patients with endometrial cancer [18].

 

Currently, whether TAPB has any effect on NLR, PLR, SII, and prognosis of endometrial cancer patients in the perioperative period is rarely reported. This study aimed to observe the effects of TAPB combined with general anesthesia on NLR, PLR, and SII in the peripheral blood of patients undergoing radical surgery for endometrial cancer, and to provide a theoretical basis for its clinical use in patients with endometrial cancer. In addition, this is the first trail to investigate the impact of TAPB on NLR, PLR and SII in patients undergoing radical endometrial cancer surgery.

## Methods and materials

### Study design and ethics

This trial was approved by the Ethics Committee of the First Affiliated Hospital of Shihezi University (KJ2023-015-01, approval date: 2023-02-10) and registered with the Chinese Clinical Trial Registration Center (ChiCTR2300072186, approval date: 2023-06-06). Written informed consent is obtained from all participants or their legal representatives prior to the commencement of any procedure.

### Participants

The participants were screened strictly according to the set inclusion and exclusion criteria. Patients were selected according to the following inclusion criteria: the pathological diagnosis was endometrial cancer; underwent elective radical operation of endometrial carcinoma under general anesthesia; aged 30–65 years; body mass index (BMI) 18–30 kg/m$^2$; American Society of Anesthesiologists class I-III; newly diagnosed patients who have not received any treatment; no distant metastasis to other parts. Patients who met any of the following criteria were excluded: skin infection at the puncture site or combined with systemic infectious diseases; associated with other malignancies, neurological disorders, endocrine system dysfunction, and hematological diseases; abnormal coagulation function; other major operations are performed within 3 months before surgery.

### Randomization and blinding

Participants who met the criteria were assigned to receive either general anesthesia (GA group) or general anesthesia combined with TAPB (TAP+GA group) in a 1:1 ratio, according to the random number table generated by IBM SPSS software version 25.0. Random numbers about the grouping were sealed in opaque envelopes and sent to the operating room. Nerve blocks were performed after induction of anesthesia by the same experienced anesthesiologist, who was not involved in other procedures of the study. The patients, researchers who performed data collection and postoperative follow-up, and clinical staff were blinded to the randomization assignments throughout the study.

### Anesthesia method

Fasting for 8 hours and abstaining from drinking for 2 hours are preoperative requirements for all participants. Once the patients enter the operating room, standardized monitoring programs should be established immediately, including electrocardiogram (ECG), heart rate (HR), non-invasive blood pressure (NIBP), and peripheral oxygen saturation (SPO2). Intravenous anesthesia induction began 3 minutes after denitrogenation and oxygen administration, followed by 0.05 mg/kg midazolam, 2 mg/kg propofol, 0.5 µg/kg sufentanil, 0.2 mg/kg cisatracurium, tracheal intubation is performed after reaching the onset time of medication and muscle relaxation. Then followed by continuous infusion of propofol at a dose of 4–8 mg/kg/h to

maintain a bispectral index (BIS) value 40–60 and remifentanil infusion rate (0.1–0.2 mg/kg/min) is adjusted to maintain mean arterial pressure (MAP) and HR within 20% of baseline values, meanwhile, cisatracurium was injected at the rate of 1–2 μg/kg/min intravenous pump to maintain muscle relaxation. Perioperative hypotension and bradycardia were treated with 40 μg deoxyadrenaline and 0.5 mg atropine, respectively. The mechanical ventilation respiratory parameters for all participants were set as follows: tidal volume 6–8 ml/kg, respiratory rate 10–14 times/min, dynamically adjusted to maintain end-tidal carbon dioxide pressure (PETCO2) between 35–45 mmHg.

Patients in the TAP+GA group disinfected the abdominal skin after induction of anesthesia, and a high-frequency line-array probe was used on a color Doppler ultrasound machine, with the probe wrapped in a luminal condom, and the ultrasound probe was scanned by placing the ultrasound probe in the mid-axillary line position between the rib margins on both sides of the abdomen and between the iliac crests, and the external abdominal oblique, internal abdominal oblique, transverses abdominus, and transversus abdominis fascia were observed ultrasonically. Under ultrasound guidance, the needle was inserted horizontally into the transversus abdominis muscle to obtain an in-plane ultrasound image of the transversus abdominis muscle, and the needle was guided to the transversus abdominis plane according to the in-plane ultrasound image, and when the tip of the needle arrived at the fascial layer between the internal oblique muscle and the transversus abdominis muscle and was withdrawn without blood or gas, 2 ml of 0.9% sodium chloride injection was injected to confirm the position of the needle. After the injection of 20 ml of 0.375% ropivacaine into each side of the abdomen, ultrasonographic observation of the diffusion of the drug and to determine whether there were any complications. TAPB was not performed in the GA group alone.

A single intravenous pre-analgesic dose of 0.1 μg/kg sufentanil was administered at the end of the procedure. Postoperative pain management was performed by routine PCIA in all patients (established with 2 μg/kg of sufentanil, 0.06 mg/kg butorphanol tartrate, and 8 mg of ondansetron in 100 mL of normal saline, and programmed with a background infusion of 2 mL/ h, 2-ml boluses and lockout interval of 15 min). When the patient experiences an unsatisfactory postoperative analgesic effect (defined as a resting VAS score $\geq$ 4), the patient is advised to independently use the analgesic pump. If the pain is not alleviated after two consecutive pumps, the attending physician will administer a single dose of 1 mg butorphanol tartrate intravenously for remedial analgesia.

## Outcome measures

The general information collection was as follows: firstly, demographic datas included age, BMI, ASA, and clinicopathologic staging of the disease; secondly, preoperative values of neutrophil, lymphocyte, platelet, NLR, PLR, and SII. (note: SII is a new observation indicator added after the ethical approval and clinical registration after a comprehensive review of a large number of literatures. It is calculated by the neutrophils, lymphocytes and platelets collected by us. Therefore, we believe that the differences of SII among groups can be further compared without further ethical review.)

Intraoperative data were sourced from the duration of anesthesia and surgery, as well as remifentanil consumption.

Postoperative observation indicators included the values of neutrophil, lymphocyte, platelet, NLR, PLR, and SII at 24 and 72 hours after surgery; VAS pain scores at 6h, 12h, 24h, and 48h postoperatively; the numbers of effective presses of the analgesic pump; the time to first ambulation and first flatus after surgery; adverse reactions of nausea and vomiting.

 

The primary outcome of this study encompassed the values of NLR, PLR, and SII. These parameters are respectively the ratio of neutrophil to lymphocyte, platelet to lymphocyte, and the ratio of the product of neutrophil and platelet to lymphocyte. Pain intensity was assessed with the Visual Analogue Scale (VAS: an 11-point scale, with 0 indicating no pain and 10 indicating the worst imaginable pain).

## Sample size calculation

In this trial, we employed PASS software version 15.0 to determine the sample size, drawing upon the postoperative 72 hours NLR levels as observed in our preliminary study. The NLR of the two groups 72 hours after operation was respectively 4.80±1.05 and 5.60±1.30. To identify a significant difference ($\alpha = 0.05$), a sample size of 36 subjects per group with a power of 80% was deemed necessary. Accounting for a dropout rate of 20%, a total of 45 subjects per group were considered for inclusion in our study.

It is worth noting that the sample size in the clinical registration scheme is based on previous research experience. After obtaining the ethical review and clinical registration consent, the sample size estimation through the pre-trial can make this study more rigorous. Therefore, the sample size in the clinical registration scheme is slightly inconsistent with the actual sample size. In addition, 72-hour postoperative NLR values were selected for sample size calculation with reference to previous studies. At the same time, SII values can be calculated from the inflammatory indicators that have been collected, so it is ethical to include it in the observation measures.

## Statistical analysis

The data were analyzed and visualized using SPSS 25.0 and GraphPad Prism 9.5.1 software. Firstly the normality of the quantitative measured data was assessed, the mean ± standard deviation (X±SD) and the median (quartile range) [M (Q1, Q3)] were used to describe data with a normally distribution and a non-normally distribution, respectively. For the primary outcome measures, the analysis of covariance(ANCOVA) was used for inter-group comparisons to control for confounding interference in the baseline values, similarly, the independent sample t test or the Mann-Whitney U test were also used to assess the diferences between groups. Categorical data were presented as proportions or percentages, with group comparison conducted using the chi-square test. No multiple comparison adjustments were applied to any of the data analyses. A *P* value of <0.05 was considered statistically significant.

## Results

### Study population

In this study, 106 patients who underwent laparoscopic radical hysterectomy for endometrial cancer were recruited at the First Affiliated Hospital of Shihezi University from June 7, 2023 to June 1, 2024. Of these, 12 patients failed to meet the inclusion criteria and 4 patients refused to participate. Subsequently, 90 participants were randomly allocated into the TAP+GA and GA groups. A total of 6 patients were not included in the final statistical analysis due to loss to follow-up, transfer to laparotomy, and so on. Consequently, 42 participants in each group completed the trial (**Fig 1**). The demographics, anesthesia, and surgical characteristics of the two groups were balanced and comparable (**Table 1**).

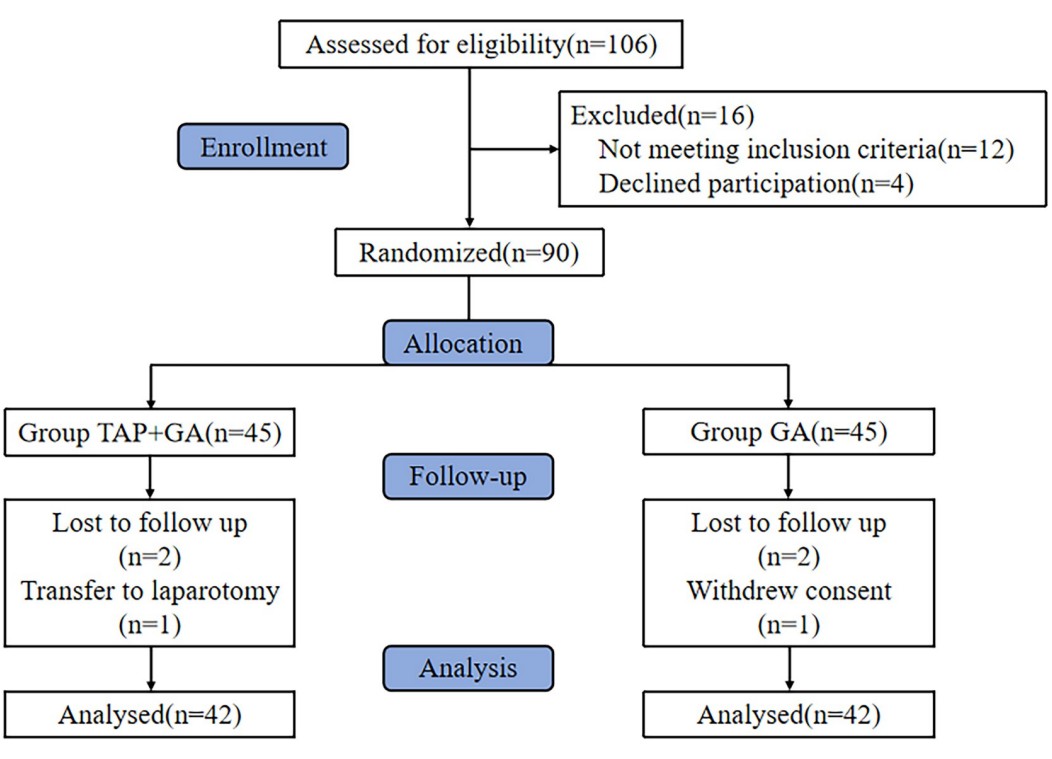

**Fig 1. The flow chart of the study.**

**Table 1. Baseline patient characteristics and surgical and anesthetic datas.**

| Parameter | Group GA (n = 42) | Group TAP+GA (n = 42) |
|---|---|---|
| Age (years) | 52.52 ±5.00 | 53.24±6.22 |
| BMI (kg/m$^2$) | 24.46±2.39 | 24.59±2.33 |
| ASA status | | |
| I | 11 (26.19%) | 15 (35.71%) |
| II | 18 (42.85%) | 16 (38.10%) |
| III | 13 (30.95%) | 11 (26.19%) |
| Clinicopathologic staging | | |
| I | 27 (64.29%) | 23 (54.76%) |
| II | 13 (30.95%) | 16 (38.10%) |
| III | 2 (4.76%) | 3 (7.14%) |
| Operation time (min) | 126.00 (118.00–131.25) | 125.50 (114.50–132.50) |
| Anesthesia time (min) | 138.00 (132.00–142.00) | 136.50 (125.25–149.25) |
| Before surgery | | |
| NLR | 1.65 (1.35–1.92) | 1.69 (1.31–2.23) |
| PLR | 137.81 (121.95–173.54) | 148.12 (124.92–191.72) |
| SII | 426.59 (337.37–588.53) | 454.62 (342.32–645.91) |

Notes: Dates are expressed as mean ± SD, or median (interquartile range), or numbers (percentage).
Abbreviations: Group GA, general anesthesia group; Group TAP+GA, transversus abdominis plane block+general anesthesia group; BMI, body mass index; ASA, American Society of Anesthesiologists; NLR, neutrophil/lymphocyte ratio; PLR, platelet/lymphocyte ratio; SII, systemic immune-inflammation index.

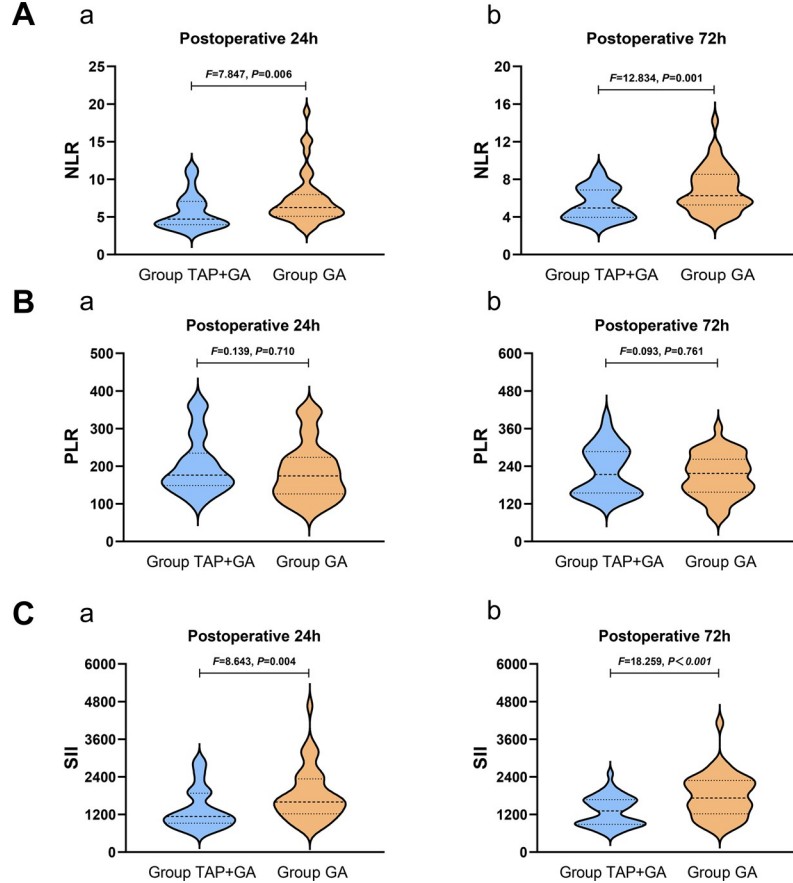

**Fig 2. Comparison of postoperative NLR, PLR, and SII between the two groups. A**: The comparison of NLR at postoperative 24h and 72h; **B**: The comparison of PLR at postoperative 24h and 72h; **C**: The comparison of SII at postoperative 24h and 72h.

**Table 2. Comparison of neutrophils, lymphocyte and platelet between the two groups.**

| Parameter | Group GA (n = 42) | Group TAP+GA (n = 42) | *P* value |
|---|---|---|---|
| **Neutrophil** | | | |
| Before surgery | 3.25±0.79 | 3.01±0.74 | 0.168 |
| 24h | 9.88±2.07 | 6.70±1.20 | <0.001 |
| 72h | 8.45±1.35 | 5.80±0.64 | <0.001 |
| **Lymphocyte** | | | |
| Before surgery | 1.90 (1.56–2.20) | 1.80 (1.50–2.03) | 0.219 |
| 24h | 1.45 (1.08–1.90) | 1.30 (0.98–1.60) | 0.075 |
| 72h | 1.20 (1.10–1.63) | 1.20 (0.90–1.40) | 0.072 |
| **Platelet** | | | |
| Before surgery | 279.69±71.32 | 282.45±60.11 | 0.848 |
| 24h | 251.14±55.37 | 243.69±56.15 | 0.542 |
| 72h | 260.14±52.99 | 243.07±51.55 | 0.138 |

Notes: Dates are expressed as mean ± SD, or median (interquartile range).

Abbreviations: Group GA, general anesthesia group; Group TAP+GA, transversus abdominis plane block+general anesthesia group.

## Primary outcomes

Fig 2 shows the comparative results of the primary indicators for two groups at different time points. It can be observed that, in comparison to the GA group, the TAP+GA group exhibited a significant reduction in NLR values at 24 and 72 hours post-surgery (4.70 [3.99,7.05] vs 6.24 [5.12,7.99], $P = 0.006$; 4.95 [3.92,6.86] vs 6.22 [5.24,8.52]), $P = 0.001$). Similarly, the values of SII at postoperative 24 h and 72 h in the TAP+GA group were significantly reduced compared to the GA group (1135.51 [922.05,1878.28] vs 1600.05 [1224.34,2335.29], $P = 0.004$; 1314.54 [886.69,1673.95] vs 1724.57 [1223.67,2285.82], $P<0.001$). And there were no significant differences in the PLR between the two groups on all time points after surgery (176.51 [148.82,234.73] vs 174.40 [126.35,223.75]), $P = 0.710$; 213.61 [154.51,286.88] vs 217.08 [157.35,262.29]), $P = 0.761$).

## Secondary outcomes

Regarding the neutrophil, TAP+GA group had a lower value than Ga group at 24 and 72 hours post-surgery (6.70±1.20 vs 9.88±2.07, $P<0.001$; 5.80±0.64 vs 8.45±1.35, $P<0.001$), and there were no significant differences in the neutrophil between the two groups before surgery ($P>0.05$). However, no significant differences were observed between the two groups before and after surgery in the values of lymphocyte, as well as the platelet ($P>0.05$) (Table 2).

Compared with the GA group, the VAS scores were found to be statistically significantly lower in the TAP+GA group at 6h, 12h, and 24 h after surgery ($P<0.001$), whereas, there were no significant differences in the VAS score at postoperative 48h postoperative 48h ($P>0.05$). Moreover, the TAP+GA group had significantly lower intraoperative remifentanil consumption and fewer numbers of postoperative analgesia pump presses compared to the GA group (1085.79±193.11 vs 1338.39±239.69, $P<0.001$; 11 (26.19%) vs 21 (50.00%), $P = 0.025$) (Table 3).

In terms of postoperative nausea and vomiting frequency, the TAP+GA group has a significant advantage over the GA group (8 (19.05%) vs 17 (40.48%), $P = 0.032$). There were no significant differences in the time to first ambulation and first flatus after surgery between the two groups ($P>0.05$) (Table 3).

**Table 3. Postoperative pain assessment indexes, nausea and vomiting frequency, and postoperative recovery of patients in two groups.**

| Parameter | Group GA (n = 42) | Group TAP+GA (n = 42) | P value |
|---|---|---|---|
| remifentanil dose (ug) | 1338.39±239.69 | 1085.79±193.11 | <0.001 |
| Postoperative VAS scores | | | |
| 6h | 2.00 (1.00–2.00) | 1.00 (0.00–1.00) | <0.001 |
| 12h | 3.00 (2.00–3.25) | 2.00 (2.00–3.00) | <0.001 |
| 24h | 3.00 (2.00–3.00) | 1.00 (0.75–2.00) | <0.001 |
| 48h | 1.00 (0.00–1.25) | 1.00 (0.00–1.00) | 0.185 |
| PONV, n% | 17 (40.48%) | 8(19.05%) | 0.032 |
| Analgesic pump compressions | 21 (50.00%) | 11 (26.19%) | 0.025 |
| Time to first ambulation (h) | 22.08±3.46 | 21.92±2.97 | 0.829 |
| Time to first flatus (h) | 24.64±5.06 | 25.08±7.54 | 0.753 |

Notes: Dates are expressed as mean ± SD, or median (interquartile range), or numbers (percentage).

Abbreviations: Group GA, general anesthesia group; Group TAP+GA, transversus abdominis plane block+general anesthesia group; VAS, visual analog scale; PONV, postoperative nausea and vomiting.

## Discussion

This double-blind, randomized controlled trial illustrated the positive effect of general anesthesia combined with ultrasound-guided TAPB in controlling postoperative inflammatory response, which was manifested by a significant decrease in the values of postoperative neutrophil, NLR, and SII. At the same time, it also reduced the consumption of opioid drugs and enhanced postoperative analgesic effects, without increasing other adverse reactions.

Surgical radical intervention for endometrial cancer stands as the most efficacious approach to address primary tumors and metastatic lesions. However, in response to surgical stimuli, the body will initiate non-specific inflammatory stress responses to maintain physiological homeostasis [19, 20]. Inflammatory responses are intimately linked to tumor genesis and progression and can impact the prognosis of cancer patients via diverse mechanisms [21, 22]. Research indicates that in endometrial cancer, inflammation enhances the expression of inflammatory cytokines via intrinsic pathways and fosters malignant tissue transformation via extrinsic pathways, thereby facilitating tumor growth [23]. These inflammatory effects can usually be assessed by blood markers such as neutrophils, lymphocytes, and platelet count, as well as NLR, PLR, and SII.

NLR and PLR respectively integrate information on neutrophils and lymphocytes, as well as platelets and lymphocytes, SII is an inflammatory marker based on counts of neutrophil, platelet, and lymphocyte, all serving as new parameters for assessing patients' immune-inflammatory system status [24, 25]. NLR, PLR, and SII have now been shown to be of significant value in the prognosis of oncological diseases [26–28]. Several meta-analyses have shown that high NLR and high PLR are significantly and strongly correlated with poorer overall survival in patients with gastrointestinal tumors and are indicative of poor disease prognosis [29, 30]. Several studies have demonstrated that a high SII serves as a reliable indicator for adverse prognoses in patients with gynecological and breast cancers, particularly in ovarian and triple-negative breast cancers [31]. Conversely, research findings suggest that patients with a low SII tend to exhibit more favorable prognoses [32]. Our trial showed that neutrophil counts, NLR, SII at 24 hours postoperatively, and centrocyte counts, NLR, SII at 72 hours postoperatively were significantly lower in the GA+TAPB group compared to the GA group. In contrast, platelet and lymphocyte counts and PLR did not show significant differences between the two groups at 24 and 72 hours postoperatively. Therefore, these results suggest that the application of combined TAPB under general anesthesia can reduce the postoperative inflammatory stress response and decrease the postoperative NLR and SII levels in patients with endometrial cancer, suggesting that TAPB may help improve the prognosis of patients with endometrial cancer.

TAPB allows local anesthetic drugs, to be injected in the plane of the transversus abdominis muscle, thus producing a blocking effect on the nerves innervating the muscles and skin of the abdominal wall [33]. Some studies have confirmed the ability of local anesthetics to inhibit the inflammatory response, thus protecting the cellular immune function of the body [34]. In this experiment, it can be observed that the number of postoperative neutrophils in the GA+TAPB group was significantly higher than that in the GA group, and the levels of PLR and SII were significantly lower than that in the GA group. Surgical stress can cause a systemic inflammatory response, leading to abnormal numbers of inflammatory factors such as neutrophils. Neutrophilia can activate the release of a variety of cytokines and molecular mediators that specifically support the process of tumor metastasis initiation, making it easier for cancer cells to be planted and metastasized [35]. Therefore, we believe that TAPB in this study can improve postoperative immune function by reducing the number of postoperative neutrophils and attenuating the postoperative inflammatory response, which in turn induces a decrease in the levels of PLR and SII.

Pain is an unpleasant emotional experience and inadequate postoperative analgesia can lead to numerous adverse events [36]. TAPB has been shown to play a positive role in the control of postoperative pain after laparoscopic hysterectomy and other abdominal surgeries [11, 37, 38]. Similar to the above studies, our results showed that patients in the TAP+GA group had significantly less intraoperative remifentanil dosage, significantly lower VAS pain scores, and significantly fewer analgesic pump presses at 6, 12, and 24 h postoperatively compared to the GA group, demonstrating that TAPB enhances postoperative analgesia in patients undergoing radical surgery for endometrial cancer, and reduces the use of perioperative opioids. Previous studies have shown that opioids are a double-edged sword, which, in addition to providing a powerful analgesic effect, can indirectly reduce natural killer cell (NK) function and lymphocyte proliferative activity, inhibit inflammatory cytokines, and then suppress immune function [39]. Therefore, we believe that TAPB in this study indirectly improved the immune function to a certain extent by reducing the pain response and the use of opioids. Meanwhile, the number of occurrences of postoperative nausea and vomiting was reduced in patients in the TAP+GA group in this study, which we believe is also related to the reduction of analgesic dosage by TAPB.

To our knowledge, our study firstly explored the effect of TAPB on inflammatory response indicators in patients with endometrial cancer after surgery, so as to estimate the effects of TAPB on tumor metastasis and prognosis in patients with cancer after surgery. Additionally, we controlled for the confounding effects of baseline NLR,PLR, and SII on the results, making the final results more convincing. During the trial, we found that ultrasound-guided visualization almost did not increase the risk of complications such as hematoma or infection that may be associated with nerve block. Our results showed that TAPB reduced postoperative NLR and PLR in patients with endometrial cancer, opening up more perspectives for the application of TAPB under the ERAS concept.

This study still has some limitations. Firstly, the trial was a single-center, small-sample size study, and the generalizability and robustness of the results are limited by the size of the sample, and further studies would need to increase the number of study participants and the type of disease. And, after preliminary estimates of changes in postoperative inflammatory markers between the two groups, we did not make multiple comparisons of NLR, PLR, and SII at different time points in the groups, considering that the trends in the two groups might not reflect the advantages of the intervention. In addition, we did not compare the effects of different access routes of TAPB on postoperative NLR, PLR, and SII in patients undergoing radical endometrial cancer surgery. Finally, long-term follow-up of the prognosis of oncology patients is also a worthwhile part that should be focused on in future studies.

## Conclusions

Ultrasound-guided TAPB reduced postoperative inflammation by lowering the levels of neutrophil, NLR, and SII, decreasing opioid consumption, and alleviating postoperative pain response, thereby affecting the prognosis of cancer patients.

## Supporting information

**S1 Checklist. CONSORT 2010 checklist of information to include when reporting a randomised trial\*.**
(DOC)

**S1 File.**
(XLSX)

**S2 File.**
(DOCX)

## Acknowledgments

We sincerely thank all participants who agreed to be included in our trial and the researchers who contributed to every aspect of the study.

## Author Contributions

**Conceptualization:** Changxu Wang, Shiwen Fan, Liping Xie, Hong Zhang.

**Data curation:** Dengfeng Gu, Jiaojiao Deng, Baobao Ma, Liping Xie, Hong Zhang.

**Investigation:** Changxu Wang, Shiwen Fan, Baobao Ma.

**Methodology:** Liping Xie, Hong Zhang.

**Project administration:** Hong Zhang.

**Resources:** Liping Xie, Hong Zhang.

**Software:** Changxu Wang, Shiwen Fan, Dengfeng Gu, Jiaojiao Deng, Baobao Ma, Liping Xie, Hong Zhang.

**Supervision:** Hong Zhang.

**Writing – original draft:** Changxu Wang, Shiwen Fan, Dengfeng Gu.

**Writing – review & editing:** Liping Xie, Hong Zhang.

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
