## [Decision Letter · Decision Letter 0]

2 Sep 2024

PONE-D-24-32923Effect of ultrasound-guided Transverse Abdominal Plane Block on NLR, PLR, and SII in patients undergoing radical resection of endometrial carcinoma: A randomized controlled trialPLOS ONE

Dear Dr. zhang,

Thank you for submitting your manuscript to PLOS ONE. After careful consideration, we feel that it has merit but does not fully meet PLOS ONE’s publication criteria as it currently stands. Therefore, we invite you to submit a revised version of the manuscript that addresses the points raised during the review process.

We look forward to receiving your revised manuscript.

Kind regards,

Aleksandra Klisic

Academic Editor

PLOS ONE

2. At PRTC, please ask the authors to include an explanation for the retrospective CT registration and confirmation that all related CTs are registered, using send back in ITC desk notes. At RTC, please check the authors' response and ping me if the authors do not address this.

3. In the online submission form, you indicated that [The data employed in the present study may be obtained from the corresponding author upon making a reasonable request.]. 

Reviewers' comments:

Reviewer's Responses to Questions

**Comments to the Author**

1. Is the manuscript technically sound, and do the data support the conclusions?

Reviewer #1: Partly

Reviewer #2: Yes

2. Has the statistical analysis been performed appropriately and rigorously? 

Reviewer #1: No

Reviewer #2: Yes

3. Have the authors made all data underlying the findings in their manuscript fully available?

Reviewer #1: No

Reviewer #2: No

4. Is the manuscript presented in an intelligible fashion and written in standard English?

Reviewer #1: Yes

Reviewer #2: Yes

5. Review Comments to the Author

Reviewer #1: This trial aimed to investigate the impact of ultrasound-guided transverse abdominal plane block (TAPB) on the systemic immune-inflammatory index (SII), peripheral blood neutrophil-to-lymphocyte ratio (NLR), and platelet-to-lymphocyte ratio (PLR) in patients undergoing radical resection of endometrial carcinoma. Patients were randomized to receive either ultrasound-guided TAPB combined with general anesthesia (GA) or GA alone. The primary outcomes were the NLR, PLR, and SII values, measured at 24 and 72 hours postoperatively. Additional outcomes included neutrophil, lymphocyte, and platelet counts; the number of effective presses on the analgesic pump; postoperative pain intensity; remifentanil consumption; and adverse reactions.

Although statistical considerations were discussed in the paper, several significant concerns have been identified:

Major Critiques:

1. Inconsistency in the Proposed Sample Size: The proposed sample size registered in the Chinese Clinical Trial Registry (www.chictr.org) was 50 patients per group. However, the Sample Size Calculation section of the paper (Lines 201–208) states a sample size of 45 patients per group. Please provide a justification for this discrepancy.

2. Primary Endpoints and Sample Size Calculation: The paper identifies three primary endpoints (NLR, PLR, and SII) measured at two time points (24 and 72 hours). However, the sample size calculation was based only on one primary endpoint (NLR) at a single time point (72 hours). Please justify this approach.

3. Lack of Multiple Comparisons Adjustment: The study reported p-values for multiple primary endpoints across multiple time points without applying any multiple comparisons adjustment. The authors should either clearly state that no multiple comparisons adjustment was applied (and include this in the limitations of the study) or apply an appropriate adjustment method, such as the Bonferroni correction.

4. Statistical Methodology: Instead of using the t-test, please apply the Analysis of Covariance (ANCOVA) method to estimate treatment effects. ANCOVA is more appropriate for adjusting baseline values of NLR, PLR, and SII.

5. Table 1: Please remove all p-values from Table 1.

6. Data Visualization: Replace the Box plots with Violin plots for a more detailed visualization of the data distribution.

Reviewer #2: Wang et al. have performed an RCT on the effect of the ultrasound-guided transverse abdominal plane block on NLR, PLR, and SII in radical resection of endometrial carcinoma. The study findings are interesting and the manuscript is well-written. These are my comments:

- The authors are suggested to use the full form of NLR, PLR, and SII abbreviations in the title of the manuscript.

- The introduction section is rather long in its current format. The authors should focus on the main ideas related to the topic and try to emphasize the gaps in knowledge.

- Authors should add strengths to their manuscript before the limitation.

- A paragraph summarizing the clinical take-home message of this manuscript should be added to the discussion.

- The references prior to 2010 could be updated with those after 2010 since they provide more up-to-date findings.

6. PLOS authors have the option to publish the peer review history of their article (what does this mean?). If published, this will include your full peer review and any attached files.

Reviewer #1: No

Reviewer #2: No

---

## [Author Response · Author response to Decision Letter 0]

28 Sep 2024

Dear Editor,

Re: Submission of the revised manuscript entitled: “Effect of ultrasound-guided Transverse Abdominal Plane Block on NLR, PLR, and SII in patients undergoing radical resection of endometrial carcinoma: A randomized controlled trial” (Submission ID PONE-D-24-32923).

Thank you very much for your letter and the reviewer’s comments which we found very helpful. We have carefully followed the comments and revised our manuscript as suggested. Enclosed below are the point-by-point responses to the reviewer’s comments (reviews’ comments in blue, our replies in black). The revised manuscript was marked with red color.

We hope the revised manuscript is satisfactory and acceptable for publication in PLOS One. Please let us know if further revisions are needed. 

We look forward to your response regarding the revised manuscript.

Best regards,

Yours sincerely 

Hong Zhang.

Department of Anesthesiology 

First Affiliated Hospital, School of Medical, Shihezi University,

Shihezi 832002, Xinjiang,

P. R. China.

Phone: +86-152-9993-5388

E-mail: 1579927013@qq.com

Specific responses to the comments:

First, we appreciate the reviewers’ valuable comments that enabled us to improve the manuscript. In the revised version, we have corrected the mistakes and inaccurate descriptions, as well as revised the entire manuscript, according to the reviewers’ comments. For clear identification, we have changes in red in the “Revised manuscript (marked-up copy)”.

Response to Reviewer 1

This trial aimed to investigate the impact of ultrasound-guided transverse abdominal plane block (TAPB) on the systemic immune-inflammatory index (SII), peripheral blood neutrophil-to-lymphocyte ratio (NLR), and platelet-to-lymphocyte ratio (PLR) in patients undergoing radical resection of endometrial carcinoma. Patients were randomized to receive either ultrasound-guided TAPB combined with general anesthesia (GA) or GA alone. The primary outcomes were the NLR, PLR, and SII values, measured at 24 and 72 hours postoperatively. Additional outcomes included neutrophil, lymphocyte, and platelet counts; the number of effective presses on the analgesic pump; postoperative pain intensity; remifentanil consumption; and adversereactions.

Although statistical considerations were discussed in the paper, several significant concerns have been identified:

Major Critiques:

1. Inconsistency in the Proposed Sample Size: The proposed sample size registered in the Chinese Clinical Trial Registry (www.chictr.org) was 50 patients per group. However, the Sample Size Calculation section of the paper (Lines 201–208) states a sample size of 45 patients per group. Please provide a justification for this discrepancy.

Response: First of all, thanks so much for your review. Your suggestion really means a lot to us. The following is our explanation of this problem, hope to get your approval. Our understanding is that ethical approval and clinical trial registration take precedence over pre-trial and formal trial. Therefore, the sample size for clinical registration was determined based on previous similar studies. With reference to the study of TAPB's influence on NLR and PLR of postoperative patients with gastric cancer, we initially set the sample size at 50 cases per group (DOI:10.3760/cma.j.cn321761-20220214-00540). The preliminary trial conducted after ethical approval and clinical registration concluded that each sample size needed to include 45 cases. For the above reasons, there is a difference in sample size between the clinical registration website and the official trial. 

2. Primary Endpoints and Sample Size Calculation: The paper identifies three primary endpoints (NLR, PLR, and SII) measured at two time points (24 and 72 hours). However, the sample size calculation was based only on one primary endpoint (NLR) at a single time point (72 hours). Please justify this approach.

Response: I'd like to thank the reviewer for precious advice. For my article, this question is very important. The reasons for this problem are explained as follows：The observational indicators considered in the initial design of this trial were NLR and PLR values, and SII was included after consulting relevant literature at the beginning of the trial, and its value could be calculated based on neutrophil, lymphocyte and platelet counts collected during the initial design, without the need to obtain additional patient data, so it was ethical. At the same time, our study believed that the intervention in our study had a positive effect as long as one of the values of NLR or PLR showed statistical significance. Therefore, referring to previous studies (DOI:10.3760/cma.j.cn321761-20220214-00540), it was pointed out that NLR had statistical significance 72 hours after surgery, while no statistical significance 24 hours after surgery, so we initially used the NLR value 72 hours after surgery to calculate the sample size.

3. Lack of Multiple Comparisons Adjustment: The study reported p-values for multiple primary endpoints across multiple time points without applying any multiple comparisons adjustment. The authors should either clearly state that no multiple comparisons adjustment was applied (and include this in the limitations of the study) or apply an appropriate adjustment method, such as the Bonferroni correction.

Response: First of all, Thank you very much for your careful review. Unfortunately, we did not conduct any multiple comparisons in this study. Our main purpose was to use covariance analysis to compare NLR, PLR, and SII at 24 hours and 72 hours after surgery between the experimental group and the control group, and to evaluate whether the intervention had an advantage based on the results. As for the values of the time points before and after the group, the preliminary estimation of the change trend of the two groups was similar, which could not be used as evidence to judge the merits of the intervention, so multiple comparisons were not made. We have explained the limitations of this manuscript. In future clinical studies, multiple comparisons will be included in statistical analyses. I have described the relevant limitations below:

And, after preliminary estimates of changes in postoperative inflammatory markers between the two groups, we did not make multiple comparisons of NLR, PLR, and SII at different time points in the groups, considering that the trends in the two groups might not reflect the advantages of the intervention. (page 21, line 377-381)

4. Statistical Methodology: Instead of using the t-test, please apply the Analysis of Covariance (ANCOVA) method to estimate treatment effects. ANCOVA is more appropriate for adjusting baseline values of NLR, PLR, and SII.

Response: We feel great thanks for your professional review work on our article. We have carried out the relevant knowledge of ANCOVA according to your suggestion, and believe that for the statistical analysis of data between the two groups, the preoperative baseline values of NLR, PLR and SII should be controlled first, which is helpful to exclude the interference and influence of the confounding factor of baseline index, so as to accurately obtain the experimental effect of intervention factors. We have performed ANCOVA of post-operative NLR, PLR and SII and described the statistical analysis results in the manuscript, which is described as follows:

Fig 2 shows the comparative results of the primary indicators for two groups at different time points. It can be observed that, in comparison to the GA group, the TAP+GA group exhibited a significant reduction in NLR values at 24 and 72 hours post-surgery (4.70 [3.99,7.05] vs 6.24 [5.12,7.99], P=0.006; 4.95 [3.92,6.86] vs 6.22 [5.24,8.52]), P=0.001). Similarly, the values of SII at postoperative 24 h and 72 h in the TAP+GA group were significantly reduced compared to the GA group (1135.51 [922.05,1878.28] vs 1600.05 [1224.34,2335.29], P=0.004; 1314.54 [886.69,1673.95] vs 1724.57 [1223.67,2285.82], P＜0.001). And there were no significant differences in the PLR between the two groups on all time points after surgery (176.51 [148.82,234.73] vs 174.40 [126.35,223.75]), P=0.710; 213.61 [154.51,286.88] vs 217.08 [157.35,262.29]), P=0.761). (page 13, line 237-248)

5. Table 1: Please remove all p-values from Table 1.

Response: Thanks for your question. We have removed the P-values in Table 1 following expert advice. The revised Table 1 is as follows:(page 12-13, line 230)

Parameter Group GA (n=42) Group TAP+GA (n=42)

Age (years) 52.52 ±5.00 53.24±6.22

BMI (kg/m2 ) 24.46±2.39 24.59±2.33

ASA status 

I 11 (26.19%) 15 (35.71%)

II 18 (42.85%) 16 (38.10%)

Ⅲ 13 (30.95%) 11 (26.19%)

Clinicopathologic staging 

I 27 (64.29%) 23 (54.76%)

II 13 (30.95%) 16 (38.10%)

Ⅲ 2 (4.76%) 3 (7.14%)

Operation time (min) 126.00 (118.00-131.25) 125.50 (114.50-132.50)

Anesthesia time (min) 138.00 (132.00-142.00) 136.50 (125.25-149.25)

Before surgery 

NLR 1.65 (1.35-1.92) 1.69 (1.31-2.23)

PLR 137.81 (121.95-173.54) 148.12 (124.92-191.72)

SII 426.59 (337.37-588.53) 454.62 (342.32-645.91)

6. Data Visualization: Replace the Box plots with Violin plots for a more detailed visualization of the data distribution.

Response: Thank you for your nice comments on our article. We have converted the Box plots in Figure 2 to a Violin plots. The new Figure 2 has been modified as follows:

Response to Reviewer2

Wang et al. have performed an RCT on the effect of the ultrasound-guided transverse abdominal plane block on NLR, PLR, and SII in radical resection of endometrial carcinoma. The study findings are interesting and the manuscript is well-written. These are my comments:

- The authors are suggested to use the full form of NLR, PLR, and SII abbreviations in the title of the manuscript.

- The introduction section is rather long in its current format. The authors should focus on the main ideas related to the topic and try to emphasize the gaps in knowledge.

- Authors should add strengths to their manuscript before the limitation.

- A paragraph summarizing the clinical take-home message of this manuscript should be added to the discussion.

- The references prior to 2010 could be updated with those after 2010 since they provide more up-to-date findings.

Response: We would like to thank the reviewers for careful and thorough reading of this manuscript and for the thoughtful comments and constructive suggestions, which help to improve the quality of this manuscript. 

First, we have adopted the suggestion of experts to make a more perfect modification of the title, and the complete title is “Effect of ultrasound-guided transverse abdominal plane block on neutrophil-to-lymphocyte ratio, platelet-to-lymphocyte ratio, and systemic immune inflammation index in patients undergoing radical resection of endometrial carcinoma”. (page 1, line 1-5)

Second, Changes have been made to the introduction section you suggested, we simplified some of the lengthy language, modified the relevant unreasonable statements, and adjusted the order of some languages to make the content more logical and better reflect the rationality of this study. We have marked them in red in the manuscript. 

the significant increase in its incidence [2, 3] and the devastating prognosis after metastasis and recurrence [4] are currently major concerns. Surgery stands as the primary method for treating endometrial cancer, multiple factors [5-8] such as anesthesia, immune system, blood transfusion, surgery, and postoperative pain can promote or hinder the metastasis of residual tumor cells in the perioperative period. Kim et al. found that the selection of different anesthesia methods during the perioperative period can directly or indirectly impact the prognosis of cancer patients [9, 10]. (page 3-4, line 52-59)

Transverse abdominal muscle plane block (TAPB) is a nerve block method, that is mostly used to relieve postoperative pain response in patients undergoing lower abdominal surgery, and it has also been found to have a positive effect on inhibiting inflammatory response and improving postoperative immune function [11]. (page 4, line 63-67)

Excessive stress on the body due to surgical trauma, extensive opioid use, and postoperative pain, all of which can suppress cellular immune function [13]. (page 4, line 71-73)

Currently, whether TAPB has any effect on NLR, PLR, SII, and prognosis of endometrial cancer patients in the perioperative period is rarely reported. (page 5, line 87-88)

In addition, this is the first trail to investigate the impact of TAPB on NLR, PLR and SII in patients undergoing radical endometrial cancer surgery. (page 5, line 92-94)

Third, we appreciate the reviewers’valuable comments that enabled us to improve the manuscript. We summarized the advantages of this manuscript and the clinical take-home message in carrying out this trial, and added them to the discussion section. It mainly includes the following points:

To our knowledge, our study firstly explored the effect of TAPB on inflammatory response indicators in patients with endometrial cancer after surgery, so as to estimate the effects of TAPB on tumor metastasis and prognosis in patients with cancer after surgery. Additionally, we controlled for the confounding effects of baseline NLR,PLR, and SII on the results, making the final results more convincing. During the trial, we found that ultrasound-guided visualization almost did not increase the risk of complications such as hematoma or infection that may be associated with nerve block. Our results showed that TAPB reduced postoperative NLR and PLR in patients with endometrial cancer, opening up more perspectives for the application of TAPB under the ERAS concept. (page 20-21, line 363-373)

 Last, we have updated the references before 2010 in the manuscript, please check in the reference section. (page 22-25, line 408-506)

For the whole article, we did make a lot of mistakes, before this, we are not serious enough, not carefully revised. Here, I would like to thank the reviewer for your careful review and point out my mistakes and problems. I hope you can give me a chance to publish this research. Here is my sincere apology and sincere gratitude.

---

## [Decision Letter · Decision Letter 1]

4 Nov 2024

PONE-D-24-32923R1Effect of ultrasound-guided transverse abdominal plane block on neutrophil-to-lymphocyte ratio , platelet-to-lymphocyte ratio, and systemic immune inflammation index in patients undergoing radical resection of endometrial carcinomaPLOS ONE

Dear Dr. zhang,

Thank you for submitting your manuscript to PLOS ONE. After careful consideration, we feel that it has merit but does not fully meet PLOS ONE’s publication criteria as it currently stands. Therefore, we invite you to submit a revised version of the manuscript that addresses the points raised during the review process.

We look forward to receiving your revised manuscript.

Kind regards,

Aleksandra Klisic

Academic Editor

PLOS ONE

Journal Requirements:

Reviewers' comments:

Reviewer's Responses to Questions

**Comments to the Author**

1. If the authors have adequately addressed your comments raised in a previous round of review and you feel that this manuscript is now acceptable for publication, you may indicate that here to bypass the “Comments to the Author” section, enter your conflict of interest statement in the “Confidential to Editor” section, and submit your "Accept" recommendation.

Reviewer #1: (No Response)

Reviewer #2: All comments have been addressed

2. Is the manuscript technically sound, and do the data support the conclusions?

Reviewer #1: Yes

Reviewer #2: (No Response)

3. Has the statistical analysis been performed appropriately and rigorously? 

Reviewer #1: Yes

Reviewer #2: (No Response)

4. Have the authors made all data underlying the findings in their manuscript fully available?

Reviewer #1: Yes

Reviewer #2: (No Response)

5. Is the manuscript presented in an intelligible fashion and written in standard English?

Reviewer #1: Yes

Reviewer #2: (No Response)

6. Review Comments to the Author

Reviewer #1: The authors have responded well to all the statistical concerns raised in the previous review. However, these explanations should be included in the revised manuscript.

Review:

1. The authors should include a brief paragraph in the revised manuscript addressing their responses to the previous critiques regarding sample size estimation (specifically critiques 1 and 2 from reviewer #1).

2. In the revised Statistical Analysis section, clearly state: "No multiple comparison adjustments were applied to any of the data analyses."

Reviewer #2: (No Response)

7. PLOS authors have the option to publish the peer review history of their article (what does this mean?). If published, this will include your full peer review and any attached files.

Reviewer #1: No

Reviewer #2: No

---

## [Author Response · Author response to Decision Letter 1]

4 Nov 2024

Specific responses to the comments:

First, we appreciate the reviewers’ valuable comments that enabled us to improve the manuscript. In the revised version, we have corrected the mistakes and inaccurate descriptions, as well as revised the entire manuscript, according to the reviewers’ comments. For clear identification, we have changes in red in the “Revised manuscript (marked-up copy)”.

Response to Reviewer

Reviewer #1: The authors have responded well to all the statistical concerns raised in the previous review. However, these explanations should be included in the revised manuscript.

Review:

1. The authors should include a brief paragraph in the revised manuscript addressing their responses to the previous critiques regarding sample size estimation (specifically critiques 1 and 2 from reviewer #1).

Response: Thank you very much for your earnest and responsible suggestion, we have added the explanation of the sample size estimate in the corresponding section. The specific expression in the manuscript is: It is worth noting that the sample size in the clinical registration scheme is based on previous research experience. After obtaining the ethical review and clinical registration consent, the sample size estimation through the pre-trial can make this study more rigorous. Therefore, the sample size in the clinical registration scheme is slightly inconsistent with the actual sample size. In addition, 72-hour postoperative NLR values were selected for sample size calculation with reference to previous studies. At the same time, SII values can be calculated from the inflammatory indicators that have been collected, so it is ethical to include it in the observation measures. (page 11, line 206-214)

2. In the revised Statistical Analysis section, clearly state: "No multiple comparison adjustments were applied to any of the data analyses."

Response: We feel great thanks for your professional review work on our article again. We have added the more rigorous statement you made to the section on statistical methods.(page 12, line 226-227)

Reviewer #2: (No Response)

Here, I would like to thank the reviewer for your careful review and point out my mistakes and problems. I hope you can give me a chance to publish this research. Here is my sincere gratitude.

---

## [Decision Letter · Decision Letter 2]

22 Nov 2024

Effect of ultrasound-guided transverse abdominal plane block on neutrophil-to-lymphocyte ratio , platelet-to-lymphocyte ratio, and systemic immune inflammation index in patients undergoing radical resection of endometrial carcinoma

PONE-D-24-32923R2

Dear Dr. zhang,

We’re pleased to inform you that your manuscript has been judged scientifically suitable for publication and will be formally accepted for publication once it meets all outstanding technical requirements.

Kind regards,

Aleksandra Klisic

Academic Editor

PLOS ONE

Additional Editor Comments (optional):

Reviewers' comments:

Reviewer's Responses to Questions

**Comments to the Author**

1. If the authors have adequately addressed your comments raised in a previous round of review and you feel that this manuscript is now acceptable for publication, you may indicate that here to bypass the “Comments to the Author” section, enter your conflict of interest statement in the “Confidential to Editor” section, and submit your "Accept" recommendation.

Reviewer #1: All comments have been addressed

Reviewer #2: All comments have been addressed

2. Is the manuscript technically sound, and do the data support the conclusions?

Reviewer #1: Yes

Reviewer #2: (No Response)

3. Has the statistical analysis been performed appropriately and rigorously? 

Reviewer #1: Yes

Reviewer #2: (No Response)

4. Have the authors made all data underlying the findings in their manuscript fully available?

Reviewer #1: Yes

Reviewer #2: (No Response)

5. Is the manuscript presented in an intelligible fashion and written in standard English?

Reviewer #1: Yes

Reviewer #2: (No Response)

6. Review Comments to the Author

Reviewer #1: The authors have responded well to the statistical issues raised in the previous review. There is no further statistical concern about this revised manuscript.

Reviewer #2: (No Response)

7. PLOS authors have the option to publish the peer review history of their article (what does this mean?). If published, this will include your full peer review and any attached files.

Reviewer #1: No

Reviewer #2: No

---

## [Editor Report · Acceptance letter]

26 Nov 2024

PONE-D-24-32923R2 

PLOS ONE

Dear Dr. Zhang, 

I'm pleased to inform you that your manuscript has been deemed suitable for publication in PLOS ONE. Congratulations! Your manuscript is now being handed over to our production team.

Kind regards, 

on behalf of

Dr. Aleksandra Klisic 

Academic Editor

PLOS ONE